# Effects of SNS Social Capital on E-Service Quality and Sustained Referral Intentions of E-Fitness Apparel: Comparative Body Image Satisfaction Analysis

**Changhyun Nam [1]**, **Jihyeong Son [2]** and **Jae-Gu Yu [3,*]**

1   Department of Apparel, Events, and Hospitality Management, Iowa State University, 31 Mackay Hall, 2302 Osborn Drive, Ames, IA 50011-1078, USA; cnam@iastate.edu
2   Department of Apparel, Merchandising, Design and Textiles, Washington State University, Johnson Hall Annex C19, Pullman, WA 99164-6406, USA; jihyeong.son@wsu.edu
3   Department of Sport Industry, Chungang University, Admin Office, Surim Gym, Seodong-daero, Daedock-myeon, Anseong, Seoul 17456, Korea
*   Correspondence: unlisted@cau.ac.kr

**Abstract:** Fitness apparel companies target consumers with easy access to social media (e.g., Facebook, Instagram, and Pinterest). However, fitness apparel companies have struggled to incorporate social interactivity into their marketing strategies due to a lack of knowledge about consumers' social media behaviors and different country contexts. The purpose of this study was to investigate (1) comparison of college students' body image satisfaction in both the United States (U.S.) and South Korea and (2) how their body satisfaction influences consumer communication and the sustained referral intentions of fitness apparel in social media. The findings from 1144 survey responses of U.S. and South Korean college students reveal that student body satisfaction differs between the two countries. Body-dissatisfied U.S. and South Korean students with social capital on social networking websites are directly influenced by word-of-mouth regarding online fitness apparel purchases. Furthermore, perceived e-service quality, including website design and website responsiveness, is a significant mediator in both cultures, affecting the word-of-mouth for fitness-related purchases. This study provides evidence for marketers of fitness apparel, particularly e-marketers, to consider the cultural differences in customer preferences and customer body satisfaction, so as to enhance service performance.

**Keywords:** fitness apparel; cross-cultural comparison; online shopping; social capital; e-service quality; sustained word-of-mouth

## 1. Introduction

Of late, the global sportswear market has grown as consumers increasingly focus on fitness activities (e.g., yoga, aerobics, and weight training), body shape and size, leisure, health, and well-being [1]. Accordingly, the sportswear market continues to grow globally and is estimated to reach USD 231.7 billion by 2024 [2]. The Sporting Goods Manufacturer Association categorizes sportswear into three groups: (a) active sports clothing or apparel specifically designed for active sporting and athletic pursuits; (b) sports clothing, or fitness or gym-oriented apparel; and (c) licensed sports clothing, which includes professional team or league logos, such as those found on jerseys [3]. Sportswear includes items such as footwear (e.g., running, tennis, and soccer shoes), clothing (e.g., golf wear, swimsuits, and those for snowboarding), and accessories (e.g., gloves, glasses, and socks). In sum, sportswear represents functional, comfortable, and safe clothing for sporting and athletic performance [4]. Generally, the term "sportswear" is used interchangeably with "activewear," "outdoor sportswear," and "fitness apparel"

in the sportswear market. This study focuses on fitness apparel, defined as indoor sports-oriented clothing and footwear (e.g., yoga pants or gym and training apparel) as consumers have become increasingly involved in fitness activities, well-being, and lifestyle sports as a result of an increased focus on their body shapes and sizes, rather than as a result of wanting to develop technical or tactical skills.

The unique social nature and benefits of social network sites (SNSs) are based on their ability to increase social capital and reduce loneliness; their utility enables users to enhance their mutual relationships [5]. In this way, consumers use SNSs to communicate their lifestyles within rubrics that include fitness activities, health and well-being, and body shape. Therefore, fitness apparel companies often try to attract consumers who have easy access to social media (e.g., Facebook, Instagram, and Pinterest) to their products with advertisements and communications [6]. While the global fitness market should also focus on incorporating social interactivity into their marketing strategies, they have thus far struggled to do so, due to a lack of knowledge about consumers' social media behaviors and a poor understanding of the different cultures they target [1]. Such divergences between online cultures are evident in the work of Kim, Sohn, and Choi [7], who identified different social relationships between American and South Korean college students on social networking sites (SNSs). South Korean college students place more emphasis on using social media to gain social support from existing social relationships, reflecting the unique social characteristics of the media, while American students are relatively more focused on using social media for entertainment by interacting within large and diverse online social venues.

Additionally, global fitness apparel companies are hindered by cross-cultural market variations in e-commerce—international consumers from various cultures have different viewpoints or approaches to body satisfaction (e.g., a particular culture may value weight, height, muscle, or body shape the most), making it difficult to create successful blanket marketing strategies [8]. For the sustainable growth of the global e-market, businesses must choose between a global and a local strategy. To establish a cultural strategy, it is first necessary to confirm consumers' cultural awareness. This study considers social media and body image satisfaction as the main determinants of cultural background.

No study has yet clarified how differences in body image satisfaction across cultures impact fitness apparel communication and purchases on social media. Responding to this research gap, this study compares United States (U.S.) and South Korean consumers' approaches to body shape satisfaction, focusing on how it relates to their use of SNSs and fitness apparel consumption. Related studies with non-U.S. samples or comparative studies with Asian samples only examine the relationship between body image dissatisfaction and risky appearance management behaviors, such as eating disorders and cosmetic surgeries [9,10].

SNSs constitute a meaningful platform to investigate because of their role in attracting young consumers to online apparel shopping in both Western and Eastern cultures [11]. Moreover, consumers' body satisfaction positively affects psychological well-being and leads to similar behaviors across the U.S. and South Korea [12,13]. However, young South Korean women demonstrate higher levels of body dissatisfaction and higher rates of self-comparison to celebrities than their U.S. counterparts [9]. Body dissatisfaction has been found to indirectly contribute to depressive moods by weakening self-esteem in American adolescents rather than in Korean adolescents [14]. We can thus infer that while e-shopping and social media are widely popular in both the U.S. and South Korea, the two countries have contrasting cultural backgrounds due to the different values and beliefs associated with Eastern and Western cultures.

By examining how consumers' body satisfaction differently affects online fitness wear purchases by culture, this study contributes to the fitness apparel industry's sustainable growth and interactions with its consumers on SNSs to increase its success in the international market. More specifically, this study investigates the relationship between consumers' perceived body satisfaction, perceived social capital, website design, website responsiveness, and sustained word-of-mouth (WOM) communication for purchasing fitness apparel, with a cross-cultural comparison of the U.S. and South Korea. Such a

comparative analysis may identify the data useful for both local and global marketing strategies. Therefore, the purpose of this study was to investigate (1) a comparison of college students' body image satisfaction in both the United States and South Korea and (2) how their body satisfaction influences consumer communication and the sustained referral intentions of fitness apparel in social media. We thus pose the following research questions:

**Research Question 1**: What differences exist in body image satisfaction between South Korean and American young consumers?

**Research Question 2**: How does social capital differently influence perceived service quality and sustained WOM for purchasing fitness apparel in the U.S. and South Korea in relation to the college students' body image satisfaction?

## 2. Theoretical Background

*Grouping by Body Image Satisfaction*

With the Internet emerging as a primary communication tool for young people worldwide, social media has become pivotal in emphasizing young consumers' societal appearance and, specifically, their body-related aesthetics, movement, perception, and expectations [13]. SNSs, including mobile social networking platforms, have the power to influence body shape ideals through consumers' subjective experiences, establish ideals or perceived body images and beauty, and influence consumers' degree of body dissatisfaction [15]. In other words, online social interactions with family members, friends, and other consumers are important contemporary formative factors of body satisfaction or dissatisfaction as e-consumers connect through SNSs to enhance communication among peers and within groups [16]. Therefore, social media plays a crucial role in dealing with body image concerns and body dissatisfaction [17]. This is because, like all media, it affects users' recognition norms.

Research on social media's effects indicates that both women and men can feel dissatisfied with their bodies when viewing idealized and unrealistic images of the bodies of celebrities and even those of their friends [15]. As such, social media can have a detrimental impact on body and appearance esteem. For example, SNSs are an important determinant of college-aged women's attitudes toward their bodies [18,19]. As the popularity of SNSs among young adults extends globally, SNSs offer individual and social opportunities for e-commerce—and yet, different cultures and genders respond in unique ways to symbols deployed on social media. For example, using social media to gather body image information negatively relates to body satisfaction in both the U.S. and South Korea [8]. Young adults, including North American university students, report dissatisfaction with their bodies when sociocultural factors influence body image perception through SNSs [20–22].

While the impact of cultural differences on social media use and body image has not been fully investigated, the cultural factor is important for understanding communications about and perceptions of body image. For example, Asian female students demonstrate lower body satisfaction in general and young Korean women's body satisfaction is highly influenced by media in comparison with that of young U.S. women [9,13,14]. However, the impact of lower body satisfaction on depressive moods through self-esteem is greater in U.S. teens [23]. The relationships between social media users, body satisfaction, and culture may be informed by cross-cultural psychology, which holds that culture impacts the perception and interpretation of social norms and cues [24,25]. For example, while Asian cultures, including that of South Korea, emphasize collectivism with a focus on equality, social support, and cooperation—ideals that support community [26]—Western cultures, including that of the U.S., emphasize individualism, idealizing self-governance, self-interest, and competition—ideals that discourage community [27]. Therefore, in a society where collectivism is emphasized, social perception and norms are important. These social norms are formulated within groups by mutual observation of and communication about collective behaviors and related self-definition. Accordingly, culture impacts the ways in which we communicate and perceive body image, such as our understanding of what makes a normative, idealized body standard—a figure often emphasized by the media. While in

collectivistic societies, individuals use social media to confirm and observe idealized body standards and communicate comfortably about normative body standards, in individualistic societies, individuals use social media relatively more to present their own or a diverse body image, an act that positively impacts their body satisfaction. Consequently, because the different ways of using social media and their impact on body satisfaction likely differ by culture, fitness apparel companies would do well to diversify, by culture, their communications about products closely tied to body satisfaction.

The generations fluent in social media and the Internet will have a different perception of body image. They are also used to online shopping and e-commerce, and the fitness apparel market is no exception. To establish a sustainable growth strategy in the global online apparel market, consumer segmentation is essential. This study thus discusses the online sustained growth of the sportswear market using the body image satisfaction levels of young South Korean and American consumers. More specifically, this study uses four groups for analysis, that is, satisfaction and dissatisfaction groups in each country. Additionally, cultural differences can be identified by the differences in the perceptions of the research variables among the four groups: (1) body-satisfied U.S. students, (2) body-dissatisfied U.S. students, (3) body-satisfied South Korean students, and (4) body-dissatisfied South Korean students.

## 3. Research Hypotheses

### 3.1. Social Capital in SNS

Social capital is defined as the sum of actual and potential resources embedded within, available through, and derived from the network of relationships possessed by an individual or social unit [28] and can be classified into (a) social capital bridging, which refers to small groups of homogeneous individuals with weak emotional or substantive relationships and (b) social capital bonding, which refers to individuals or groups with strong emotional or substantive relationships [29]. The social capital in SNSs reflects networks' various potential roles in online shopping. Combinations of social activities and commercials will attract consumers, while marketing and purchase stimuli have a stronger influence in SNSs due to the time saved and information obtained by e-consumers when browsing websites [30]. Moreover, SNSs create opportunities for e-consumers to communicate with friends, browse products, and read other consumers' reviews. For example, consumers interested in apparel websites can find coupons, promotions, and events and can not only search for these on SNSs and share their experiences with friends and social groups, but also spontaneously buy products. For instance, Nam, Kim, and Kwon [31] found that social capital plays an important role in SNSs' influence on purchase intentions for sustainable outdoor sportswear. Moreover, SNSs can easily and quickly provide useful information about new sportswear to young consumers.

The relationships formed through SNSs are critical for consumer socialization [32]. The positive effects of SNSs include participants generating strong social ties and frequent interactions with close friends, as well as the emotional or substantive support obtained through such intensive communication. Huang [30] demonstrates that SNSs relate to social bonding and bridging and can influence consumers' affective perceptions, which induces users to visit websites with a particularly appealing atmosphere. Moreover, SNS users share various kinds of information, including followers' feedback and personal negative or positive experiences, quickly [33]. The strength of the relationship between consumers (SNS users) and e-tailers can build social interaction ties [34]. In e-commerce, transactions between consumers and e-tailers are carried out on websites [35]. SNS users are influenced by website quality—which includes content, functions, and user involvement—which sometimes prompts them to make impulsive decisions [36,37]. Lin and Lu [38] found that social interaction is the main factor in determining sustainable use intention among SNS users (e.g., Facebook). In addition, social interaction ties have a strong and positive effect on mobile service-based WOM [39]. We thus posit the following hypotheses:

**Hypothesis 1 (H1).** *Social capital in SNSs positively influences perceived e-service quality in terms of website design.*

**Hypothesis 2 (H2).** *Social capital in SNSs positively influences perceived e-service quality in terms of website responsiveness.*

**Hypothesis 3 (H3).** *Social capital positively influences sustained WOM for fitness apparel.*

*3.2. Perceived E-Service Quality*

Social e-services refer to web-based services delivered to customers through the Internet [40]. In an e-commerce context, a variety of e-service quality attributes (e.g., website design, trust in the e-service, responsiveness, and personalization) help establish the stimuli for consumers' e-commerce shopping behaviors, although these stimuli have distinct effects [32,41–43] Perceived e-service quality refers to an e-commerce website's overall desirable performance [44]. Al-Debei, Akroush, and Ashouri [41] reported that e-commerce attributes involve e-consumers' evaluations and judgments of a website's design and processes. For example, when a customer's expectations are met or exceeded by a websites' features, the perceived quality of the e-service could be rated as "good" or even "excellent." Similarly, e-service vendors can attract consumers using their websites' visual effects, thus allowing e-consumers to experience the convenience and pleasure of online shopping.

E-commerce provides consumers with a vast assortment of information regarding products and services, along with various safeguards in the form of security features. Further, e-service quality can also improve the relationship between companies and e-customers in their online interactions. As such, online retail stores must focus on e-service quality indicators, such as website quality [45,46], security [47], website design [37,48], and navigability [49], all of which lead to enhanced online interactions for consumers. These elements of e-service quality are important in an online shopping environment, as they influence the psychological processes impacting consumers' purchase intentions [42,50] or, ultimately, WOM communication in e-commerce [41,45]. Therefore, it is important to comprehend the extent of e-shoppers' behaviors and benefits in an e-service quality context, such as ease of use, pricing, and time savings.

Additionally, perceived website service quality generates positive e-customer satisfaction, company loyalty, and the overall attitudes toward online shopping [51], as the Internet is widely used as an omni-channel by e-retailers and e-marketers. First, websites' visual effects and their design and functionality drive consumers' positive experiences. Another variable is website responsiveness, or the help and responses given to e-consumers' requests—whether for information, service, returns, or exchanges, such responses must be quick in online shopping [52]. Previous research has shown that responsiveness impacts perceived service quality and customer satisfaction in e-commerce [53,54]. Overall, e-service quality involves providing consumers with a superior experience relative to the interactive information flow [55]. In addition, Hossain and Kim [33] reported that e-service quality strongly influences WOM and the sustainable use intention of SNS users (e.g., Facebook). Our study postulates that perceived e-service quality in terms of website design and responsiveness significantly influences WOM communication in purchasing fitness apparel online. Therefore, we present the following hypotheses:

**Hypothesis 4 (H4).** *Perceived e-service quality in terms of website design influences sustained WOM for purchasing fitness apparel.*

**Hypothesis 5 (H5).** *Perceived e-service quality in terms of website responsiveness influences sustained WOM for purchasing fitness apparel.*

### 3.3. Word-of-Mouth

Levy and Gvili [16] reported that both bonding and bridging social capital contribute to the increased credibility of electronic WOM (E-WOM) communication. E-WOM communication refers to "any positive or negative statement made by potential, actual, or former customers about a product or company, which is made available to a multitude of people and institutions via the Internet" (p. 39) [56]. WOM allows consumers to consistently engage in either information about or rational standards for product assessment and the purchase intentions of current and future consumers [57]. Through the positive WOM associated with attitudes toward e-commerce, either bridging social capital (creation of weak ties) or bonding social capital (creation of strong ties) is also perceived as reliable in e-commerce [41]. Therefore, social capital's effects guide consumers' purchase intentions and interactions with WOM as well as diversified SNS users from different backgrounds that prefer receiving a variety of information in weak social connections [58]. A high-quality website positively influences SNS users' sustained WOM, and thus the users will be willing to purchase fitness apparel through the SNS's information [35]. Perceived e-service quality can be mediated by the relationships between the social capital in SNSs and sustained WOM communication for purchasing products online. Figure 1 shows all of our proposed hypotheses.

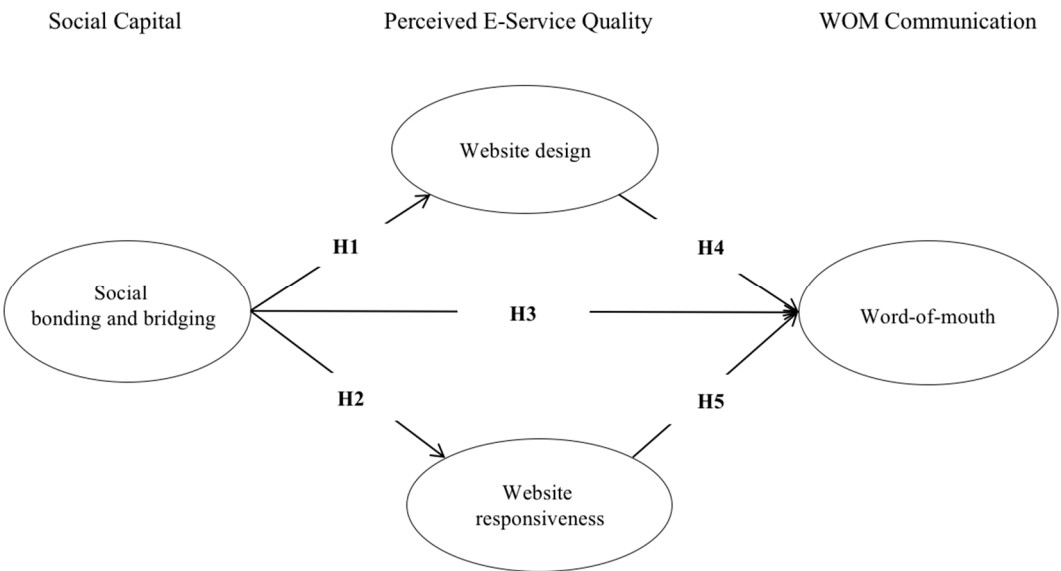

**Figure 1.** Conceptual model.

## 4. Data and Methodology

### 4.1. Subjects and Data Collection

Students attending undergraduate and graduate classes at a university located in the Southeastern U.S. and three universities in South Korea were recruited as survey participants. From the 1219 voluntary participants, a total of 1144 valid responses were obtained from the U.S. ($N = 607$) and South Korean students ($N = 537$). Most of the participants were female (81% in the U.S., 75% in South Korea). Both cultural groups used computers (88% and 57% for U.S. and South Korean students, respectively) and smartphones (9% and 42%, respectively) to purchase fitness apparel online. More than 80% of the participants in both countries spent an average of less than USD 700 annually on fitness apparel, including running shoes and accessories. Regarding their average daily use of SNSs, approximately 41% of the U.S. students spent three to six hours per day on SNSs, while 48% of the South Korean students spent one to three hours on these sites daily. As for the reported exercise frequency, 44% of the U.S. students and 55% of the South Korean students said they worked out at gyms or fitness

centers at least once every seven days. Approximately 40% of the students considered purchasing fitness apparel according to the clothing's features—quality (24%, 20%), function (23%, 21%), price (19%, 10%), and size (13%, 12%)—in the U.S. and South Korea, respectively.

### 4.2. Measures

The measurement items were adapted from previous studies to fit this study's context. To avoid linguistic bias and errors in the survey questionnaire, we constructed the survey questionnaire by first creating an English version, then translating the content from English to Korean using back-translation [59]. The participants were surveyed regarding their: (a) demographic information; (b) body satisfaction measured by cognitive attitudes toward body image such as perceived own body image (*"Choose the figure that reflects how you think you look currently?"*), ideal body image (*"Choose the figure that reflects how you would like to look?"*), and satisfied body image (*"Are you satisfied with your body?"*), and preference for losing weight (*"Would like to lose weight?"*) [60]; (c) perceptions of social capital on SNSs [61]; (d) perceptions about websites' design and responsiveness and their impacts on e-service quality [42]; and (e) perceived influence of word-of-mouth (WOM) communication [57] using a seven-point Likert-scale ranging from 1 = strongly disagree to 7 = strongly agree. The items were pre-vetted by sport management and apparel consumer behavior professors. In addition, a preliminary survey was conducted to confirm the validity and reliability of the survey instrument.

### 4.3. Validity and Reliability

Before the structural equation modeling (SEM), confirmatory factor analysis (CFA) was conducted to confirm the validity of the survey variables. The CFA and reliability results are shown in Table 1.

**Table 1.** CFA results for the measurement model.

| | Scale Items | $\beta$ | SE | CR | Cronbach's $\alpha$ |
|---|---|---|---|---|---|
| | **Social Capital (SC)** | | | | 0.89 |
| SC1 | Help solve my problems in SNSs | 0.689 | | | |
| SC2 | Meet new people in SNSs | 0.692 | 0.05 | 21.35 *** | |
| SC3 | Support online community activities | 0.785 | 0.05 | 23.92 *** | |
| SC4 | Advice (important decision) in SNSs | 0.796 | 0.05 | 24.21 *** | |
| SC5 | Good references (e-services and products) | 0.722 | 0.05 | 22.19 *** | |
| SC6 | Several people in SNSs | 0.836 | 0.05 | 25.25 *** | |
| | **Website Design (WD)** | | | | 0.86 |
| WD1 | Visually appealing | 0.703 | | | |
| WD2 | A well-organized appearance | 0.913 | 0.05 | 27.42 *** | |
| WD3 | Quick and easy to complete a transaction | 0.854 | 0.05 | 26.54 *** | |
| | **Website Responsiveness (WR)** | | | | 0.87 |
| WR1 | Quickly respond to customer requests | 0.785 | | | |
| WR2 | Always willing to help customers | 0.871 | 0.04 | 30.37 *** | |
| WR3 | Prompt service | 0.844 | 0.04 | 29.66 *** | |
| | **Sustained Word-of-Mouth (WOM)** | | | | 0.93 |
| WOM1 | Positive experience for buying fitness apparel | 0.946 | | | |
| WOM2 | Friends and families for buying fitness apparel | 0.956 | 0.02 | 62.60 *** | |
| WOM3 | Users and followers for buying fitness apparel | 0.816 | 0.02 | 41.28 *** | |
| **Model Fit Index** | | $\chi^2$ (*df*) | CFI | NFI | SRMR | TLI |
| | | 451.21 (84) | 0.97 | 0.96 | 0.04 | 0.96 |

Note: $\beta$ = estimate value, SC = social capital, WD = website design, WR = website responsiveness, WOM = word-of-mouth communication, SE = standardized estimate, CR = critical ratio, $\chi^2$ = chi-square, *df* = degree of freedom, CFI = comparative fit index, NFI = normed fit index, SRMR = standardized root mean square residual, TLI = Tucker-Lewis index, *** $p < 0.001$.

Specifically, CFA was conducted to examine the factors' structure and the measurement model's scale validity to determine the effect of social capital on WOM communication. Based on the rule of thumb criteria for goodness-of-fit [62], a reasonable minimum value for model acceptance of 0.90 indicates a good or acceptable model fit, while standardized root mean square residual (SRMR) values below 0.10 also indicate a good ($0 \leq$ SRMR $\leq 0.05$) or acceptable fit ($0.05 <$ SRMR $\leq 0.10$). Furthermore, the constructs' Cronbach's alpha coefficients ranged from 0.86 to 0.93, notably above the recommended cutoff point of 0.70 [63]; this indicates the measurement model's construct validity. As shown in Table 2, the results of measurement model fitness using maximum likelihood are acceptable ($\chi^2 = 451.2$, $df = 84$, $p < 0.001$, CFI $= 0.97$, TLI $= 0.96$, NFI $= 0.96$, SRMR $= 0.04$).

**Table 2.** Results of k-means cluster analysis for grouping.

| Types of Body Image Items | Satisfaction N = 411 (35.9%) | Dissatisfaction N = 733 (64.1%) | Mean Square | df | F-Value | Sig. |
|---|---|---|---|---|---|---|
| Body concerned (negative) | 3.00 | 4.00 | 1.14 | 1.142 | 356.51 | 0.000 *** |
| Ideal body (positive) | 3.06 | 2.79 | 1.00 | 1.142 | 19.06 | 0.000 *** |
| Satisfied body (positive) | 5.38 | 3.47 | 2.25 | 1.142 | 426.40 | 0.000 *** |
| Prefer to lose weight (negative) | 2.57 | 6.21 | 1.29 | 1.142 | 2705.57 | 0.000 *** |

Note: $df$ = degrees of freedom, sig. = two-tailed significance, *** $p < 0.001$.

### 4.4. Data Processing

The survey instrument's content validity was checked by three qualified scholars with extensive expertise in consumer behavior and marketing and construct reliability analysis was conducted using Cronbach's alpha in SPSS 23 and AMOS 23. The statistical analysis was conducted in five steps. First, the participants were classified into two groups by their body satisfaction using k-means cluster analysis and an independent t-test: body satisfaction and body dissatisfaction. Chi-square analysis was used to determine the differences among the four clusters, namely (a) body-satisfied U.S. students, (b) body-dissatisfied U.S. students, (c) body-satisfied South Korean students, and (d) body-dissatisfied South Korean students. Second, a one-way ANOVA and a Scheffe's post hoc test were used to compare the recognition characteristics of each group. Third, the conceptual model's hypotheses for all data were examined through SEM. A two-step approach was used in SEM to confirm the measurement model's overall factor structure and test the relationships between the hypothesized latent constructs. Fourth, the relationships between a set of indicators representing the measurement model's latent constructs were assessed through CFA. Given the fitness of the measurement model, SEM was then conducted for the hypothesized interrelationships among the latent constructs. Fifth, SEM analysis was performed for each of the four groups and path differences were compared. All significance levels were controlled at the $p < 0.05$ level.

## 5. Results

### 5.1. Grouping by Body Image Satisfaction

First, we examine Research Question 1, *what differences exist in body image satisfaction between South Korean and American young consumers?* As a result of the cluster analysis based on the four question items about body image satisfaction among young people in South Korea and the U.S., the subjects were classified into two groups by body image satisfaction.

As Table 2 illustrates, the participants' responses generated two clusters—body satisfaction (N = 411) and dissatisfaction (N = 733)—according to their cognitive attitudes toward body image and the preferred amount of weight to lose. Of these participants, 64% were dissatisfied with their body image and wanted to lose weight. Anderson-Fye [64] argued that the Western-style thin body ideal is not universal in non-Western cultures. However, social media's effect on shaping body image ideals is prevalent in both Western and non-Western cultures.

Table 3 shows each group's differences in the perception of the various factors. The two groups were then subdivided into four by country: (a) body-satisfied U.S. students, (b) body-dissatisfied U.S. students, (c) body-satisfied South Korean students, and (d) body-dissatisfied South Korean students. Group sizes are as follows: body satisfaction ($N_{U.S.}$ = 337, $N_{Korea}$ = 396) and dissatisfaction ($N_{U.S.}$ = 270, $N_{Korea}$ = 141), with $\chi^2$ = 41.12, *df* = 1, *p* < 0.001. College students in both cultures seek to have positive perceptions of their bodies as well as healthy lifestyles, and body satisfaction seems to be associated with improved psychological well-being [13]. Therefore, the results confirm that the differences in the perceptions of SNS social capital, service quality satisfaction, and sustained WOM explain the cultural differences between groups.

**Table 3.** Variable mean differences among the four groups.

| | Group | Mean (SD) | F-Value | Sig. | Scheffe's Post Hoc |
|---|---|---|---|---|---|
| **Social Capital** | | | 30.24 | 0.000 *** | 1 and 2 < 3 and 4 |
| South Korea | Satisfactory 1 | 4.00 (1.12) | | | |
| | Unsatisfactory 2 | 3.82 (1.12) | | | |
| United States | Satisfactory 3 | 4.58 (1.19) | | | |
| | Unsatisfactory 4 | 4.59 (1.11) | | | |
| **Website Design** | | | 153.77 | 0.000 *** | 1 and 2 < 3 and 4 |
| South Korea | Satisfactory 1 | 4.56 (0.94) | | | |
| | Unsatisfactory 2 | 4.58 (1.09) | | | |
| United States | Satisfactory 3 | 5.81 (0.85) | | | |
| | Unsatisfactory 4 | 5.69 (0.96) | | | |
| **Website Responsiveness** | | | 53.80 | 0.000 *** | 1 and 2 < 3 and 4 |
| South Korea | Satisfactory 1 | 4.14 (0.92) | | | |
| | Unsatisfactory 2 | 4.13 (1.09) | | | |
| United States | Satisfactory 3 | 4.94 (1.02) | | | |
| | Unsatisfactory 4 | 4.83 (1.05) | | | |
| **Sustained Word-of-Mouth** | | | 79.14 | 0.000 *** | 1 and 2 < 3 and 4 |
| South Korea | Satisfactory 1 | 4.29 (1.05) | | | |
| | Unsatisfactory 2 | 4.06 (1.15) | | | |
| United States | Satisfactory 3 | 5.30 (1.11) | | | |
| | Unsatisfactory 4 | 5.15 (1.20) | | | |

Note: SD = standard deviation, sig. = two-tailed significance, *** *p* < 0.001.

Further exploration of the four variables revealed statistically significant differences among the four groups for each variable: social capital (F = 30.2), website design (F = 153.8), website responsiveness (F = 53.8), and WOM communication (F = 79.1), with p-values of 0.001 (see Table 3). Body-satisfied U.S. and body-dissatisfied South Korean students—with the largest means of M = 5.81 and M = 4.58, respectively—were found to be influenced by the website design of fitness apparel brands. This suggests that e-tailers and marketers should substantially focus their marketing efforts on high-quality website design for their e-services to generate positive online WOM. Although vendors should establish different strategies in response to variations in consumer body perceptions between the two cultures, cross-cultural marketing is more likely to be effective for online fitness apparel purchases. It is important to note that, regarding the differences in approaches to body image between South Korea and the U.S., we found that the U.S. students have higher mean scores than the South Korean students in each element (social capital, website design, website responsiveness, and sustained WOM) than the South Korean students. Therefore, it is necessary to consider the possibility of generalization, that Americans have higher social capital than South Koreans, and that Americans demonstrate very active consumption practices (e.g., e-service quality cognition, and WOM). Our results were realistic along these lines.

### 5.2. Hypothesis Testing the Conceptual Model

Second, we examine Research Question 2, how does social capital differently influence perceived service quality and sustained WOM for purchasing fitness apparel in the U.S. and South Korea in relation to young students' body image satisfaction? The verification of the model involved a test for the global consumer to confirm the validity of the SEM model for all subjects as well as one for the four groups to identify the differences between their SEM paths. The SEM primarily focused on an assessment of the proposed model pertaining to the hypothesized paths among the latent constructs. As per Figure 2, the goodness-of-fit indices support the structural relationships among the latent constructs underlying the empirical data ($\chi^2$ = 694.5, df = 85, p < 0.001, CFI = 0.95, TLI = 0.93, NFI = 0.94, SRMR = 0.096).

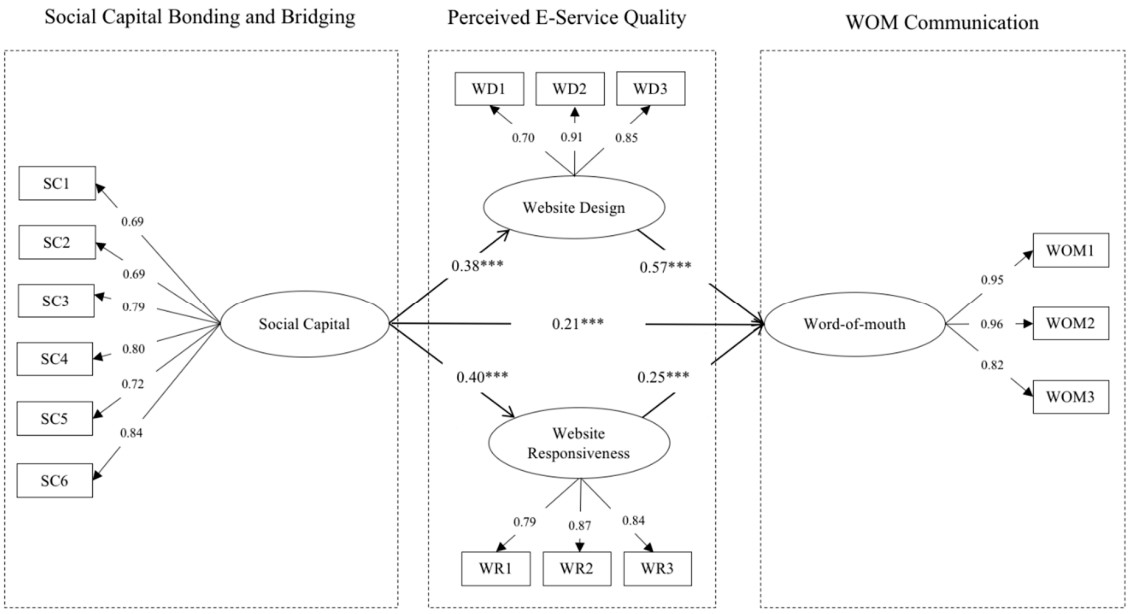

**Figure 2.** Results of the hypotheses testing for the conceptual model. Note: $\chi_2$ = 694.5, *df* = 85, standardized root mean square residual (SRMR) = 0.096, comparative fit index (CFI) = 0.95, Tucker-Lewis index (TLI) = 0.93, normed fit index (NFI) = 0.94, RMSEA = 0.096, *** *p* < 0.001.

Testing the hypotheses for the overall conceptual model revealed that social capital on SNSs significantly affected website design ($\gamma = 0.38, p < 0.001$) and website responsiveness ($\gamma = 0.40, p < 0.001$), which in turn significantly influenced WOM communication ($\gamma = 0.57$ and $\gamma = 0.25$, respectively; $p < 0.001$). Additionally, social capital's influence on WOM communication was significantly positive ($\gamma = 0.21, p < 0.001$). Therefore, e-service website design quality and responsiveness partially mediate social capital's effect on WOM communication when purchasing fitness apparel. In other words, perceived e-service quality is a significant mediator of sustained WOM communication for fitness apparel purchases in both the U.S. and South Korea, as it provides young adults with greater control in their e-commerce fitness apparel consumption. Next, the SEM path differences were analyzed for each body image satisfaction group.

Table 4 displays the results of the research questions for the four different groups. These results confirm that the goodness-of-fit indices support the structural relationships among the latent variables ($p < 0.001$). The results of H3 and H4 identified partially different outcomes among the four groups. However, website design was not found to influence WOM communication for purchasing fitness apparel among body-dissatisfied South Korean students. Further, social capital had no influence on WOM communication for both body-satisfied U.S. students and body-dissatisfied South Korean students. However, substantial differences in social media use may exist as determined by each culture's preference for social media tools (e.g., Kakao talk in South Korea versus Snapchat in the

U.S.) although Facebook, YouTube, and Instagram are globally dominant. Moreover, social capital had no direct effect on WOM communication for purchasing fitness apparel for body-satisfied U.S. and body-dissatisfied South Korean students. However, social capital was found to indirectly affect body-dissatisfied South Korean students through website responsiveness to WOM communication. Therefore, e-service quality (web responsiveness) plays an important mediating role toward enhancing social capital and WOM communication for purchasing fitness apparel in both cultures, regardless of body satisfaction.

**Table 4.** Variable means among the four groups.

| | | Satisfaction ($n$ = 337) | | Dissatisfaction ($n$ = 270) | |
|---|---|---|---|---|---|
| | **Hypotheses** | **Hypothesis Testing** | **SE (CR)** | **Hypothesis Testing** | **SE (CR)** |
| **United States** **($N$ = 607)** | H1: SC → WD | supported | 0.26 (4.10 ***) | supported | 0.27 (3.79 ***) |
| | H2: SC → WR | supported | 0.36 (5.57 ***) | supported | 0.28 (3.83 ***) |
| | H3: SC → WOM | not supported | 0.12 (1.91) | supported | 0.19 (3.00 **) |
| | H4: WD → WOM | supported | 0.34 (5.70 ***) | supported | 0.41 (6.34 *) |
| | H5: WR → WOM | supported | 0.17 (2.87 **) | supported | 0.21 (3.46 ***) |
| | **Model Fit Index** | | | | |
| | $\chi^2(df)$ | | 253.63 (85) | | 205.77 (85) |
| | $\chi^2/df$ | | 2.98 | | 2.42 |
| | TLI | | 0.93 | | 0.94 |
| | CFI | | 0.95 | | 0.95 |
| | NFI | | 0.92 | | 0.92 |
| | SRMR | | 0.09 | | 0.01 |
| | *P*-value | | 0.000 *** | | 0.000 *** |
| | | Satisfaction ($n$ = 396) | | Dissatisfaction ($n$ = 141) | |
| | **Hypotheses** | **Hypothesis Testing** | **SE (CR)** | **Hypothesis Testing** | **SE (CR)** |
| **South Korea** **($N$ = 537)** | H1: SC → WD | supported | 0.41 (5.66 ***) | supported | 0.58 (4.58 ***) |
| | H2: SC → WR | supported | 0.15 (5.92 ***) | supported | 0.48 (4.77 ***) |
| | H3: SC → WOM | supported | 0.31 (4.77 ***) | not supported | 0.18 (1.42) |
| | H4: WD → WOM | supported | 0.18 (3.62 ***) | not supported | 0.18 (1.55) |
| | H5: WR → WOM | supported | 0.15 (2.62 **) | supported | 0.20 (2.06 *) |
| | **Model Fit Index** | | | | |
| | $\chi^2(df)$ | | 356.64 (85) | | 205.26 (85) |
| | $\chi^2/df$ | | 4.20 | | 2.42 |
| | TLI | | 0.90 | | 0.89 |
| | CFI | | 0.92 | | 0.91 |
| | NFI | | 0.89 | | 0.85 |
| | SRMR | | 0.09 | | 0.09 |
| | *p*-value | | 0.000 *** | | 0.000 *** |

Note: SC = social capital, WD = website design, WR = website responsiveness, WOM = word-of-mouth communication, SE = standardized estimate, CR = critical ratio, $\chi^2$ = chi-square, $df$ = degree of freedom, TLI = Tucker-Lewis index, CFI = comparative fit index, NFI = normed fit index, SRMR = standardized root mean square residual, * $p < 0.05$, ** $p < 0.01$, and *** $p < 0.001$.

## 6. Discussion and Implication

This study analyzed the sustainability of recommendation intentions to measure the sustained growth of the e-shopping market for a better understanding of young consumers' body satisfaction. The aim was to determine whether cross-cultural marketing may enhance the success of fitness apparel e-tailers and e-marketers. Limited research has been conducted using our proposed conceptual model for measuring the association between differences in body satisfaction between the U.S. and South Korea, WOM communication, and the purchase of fitness apparel. Thus, this study is unique in its testing of this conceptual model.

Our findings prove insightful. First, significant differences exist in perceptions of and satisfaction with body image among college students in the two cultures [9,13,14]. Specifically, South Korean students were more satisfied with their bodies than U.S. students. Lee et al. [13] found that young

adults in both the U.S. and South Korea show positive relationships between social media used for information-sharing and perceived body image. However, a variation exists in the level of body satisfaction; respondents' body images depend on their involvement in SNSs. Different body perceptions and cultural aspects relate to the key features of fitness apparel—function and quality—and thus affect the intensity, style, and purpose of WOM communication about fitness apparel available for purchase online. Ultimately, these results provide insights for optimizing marketing strategy to assist the global fitness apparel industry, including segmentation approaches to advertising that involve targeting the norms of body satisfaction and fitness apparel consumption of a particular culture. It is imperative to the industry's success that marketers identify strategies that respond to both cultural approaches to the body and the features of their fitness apparel.

Second, positive relationships exist among the conceptual model's variables for purchasing fitness apparel, including relationships between social capital on SNSs, website design and responsiveness for perceived e-service quality, and WOM communication among both U.S. and South Korean students. One key finding is that both cultures positively associate SNS social capital with their perception of the related website's service quality and WOM communication when purchasing fitness apparel online. Additionally, website design and responsiveness were shown to mediate the relationship between SNS social capital and WOM communication. Consequently, perceived e-service quality (web responsiveness) plays a significant role as a mediator in shopping for fitness apparel online. Moreover, the literature revealed that U.S. consumers' perceptions and attitudes are more highly determined by internal attributes and are relatively free from the external influence of others' opinions in terms of purchase experience [65]. By contrast, South Korean students satisfied with their body images are influenced by social capital and tend to engage in more WOM communication than those who are unsatisfied. Similarly, Lee et al. [13] found that South Koreans are concerned with how others perceive them and require others' approval of their body images through social media. Thus, social media more strongly influences body satisfaction in Korean college students than in U.S. college students.

Third, website design did not influence WOM communication among body-dissatisfied South Korean students. Therefore, e-tailers and marketers should enhance their website designs to optimize online fitness apparel purchasing behaviors. Other website service quality attributes, such as trust, reliability, and personalization, can potentially bridge the gap between different cultures and body image satisfaction in online fitness apparel shopping.

Finally, SNS social capital was not found to influence WOM communication for purchasing fitness apparel in both body-satisfied U.S. and body-dissatisfied South Korean students. Therefore, it is important for e-marketers to comprehend not only the effects of culture and SNS social capital, but also how these effects influence young consumers' WOM communication in purchasing fitness apparel. We suggest that fitness apparel e-tailers continue to develop and improve website design and communicate with consumers to provide better information regarding fitness apparel and consequently increase e-consumers' WOM communication. In short, successful marketing strategies in the fitness apparel market require the identification of target consumers and a sense of what marketing mix will best satisfy their needs. Based on the participants' responses regarding social media interactions, we assume that online fitness apparel stores have already started to improve their communication and shared social capital with consumers in terms of both bonding and bridging, and that this has, in turn, increased sales.

What is also important to take away from this is that our conceptual model serves to identify the effect of social capital on sustained WOM. Moreover, our results reveal that social capital influences sustained referral intention in social media and e-marketing, with online service quality a mediating factor. In other words, social capital impacts e-service quality perception, and perceived e-service quality perception impacts WOM. Accordingly, we suggest that, in order to sustainably grow, online retailers would do well to link SNS with their homepages for reviews and recommendations—any

good recommendations of a product strengthen its producer's social capital [66], a principle that holds true online as well [67].

## 7. Limitations and Future Research

It is important to note that our sample included only college students and that more female than male students participated from both the U.S. and South Korean populations. As such, future studies using the proposed conceptual model should consider balanced samples in terms of gender and a broad range of ages. Another limitation is the study's geographical context, in that it involves single areas in both the U.S. and South Korea. Repeating the analysis in different states or countries will most likely strengthen and validate the findings. We also recommend that additional research compares bridging and bonding in SNS social capital and their effects on sustained WOM communication, because a marketing segmentation strategy can divide target consumers into subgroups in e-commerce. This investigation can be expanded to include other variables, such as attributes of perceived e-service quality, to determine fitness apparel online purchase intentions among different cultures. However, this study provides information for the establishment of global and subcultural strategies, thus contributing to the sustainable growth of the sportswear e-market. Furthermore, it examines the concept of a cultural strategy for the global market.

**Author Contributions:** Investigation, C.N.; Conceptualization, J.-G.Y. and C.N.; Writing—Original Draft Preparation, C.N.; Methodology, J.-G.Y. and J.S.; Software, J.-G.Y.; Validation, J.-G.Y. and J.S.; Formal Analysis, J.-G.Y.; Data Curation, J.-G.Y.; Writing—Review and Editing, C.N. and J.S.; Visualization, C.N. and Y.J; Project Administration, C.N; Resources, C.N.; Supervision, J.-G.Y.

**Funding:** This research received no external funding.

**Conflicts of Interest:** The authors declare no conflict of interest.

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
