# Peer review of "Effects of SNS Social Capital on E-Service Quality and Sustained Referral Intentions of E-Fitness Apparel: Comparative Body Image Satisfaction Analysis"

_sustainability, doi:10.3390/su11247154_

Round 1
Reviewer 1 Report
Please see the attached.

Author Response
R1 Comment 1: The abstract needs to be completely rewritten. It is vital to write a complete but concise description of your work to entice potential readers into obtaining a copy
of the full paper. Possibly follow a checklist consisting of: motivation, problem
statement, approach, data, results, and conclusions.
Authors’ Response: Thank you for your insightful suggestion and feedback. Based on your suggestion, we made a change in the abstract. [see P. 1. Line 14-27]
R1 Comment 2: The arguments presented do not flow logically. The author(s) might consider
revising the paper presentation to: 1. Introduction 2. Theoretical Background 3.
Research Hypotheses 4. Data and Methodology 5. Results 6. Discussion and Implications 7.
Limitations and Future Research.
Authors’ Response: We thoroughly revised each section according to your insight comments.
R1 Comment 3: The introduction should be completely rewritten. Introductions are important.
They arouse a reader's interest, introduce the subject, and tackle the So What? factor. Possibly considering the following approach can produce an effective introduction. Describe the importance of the study - why was this worth doing in the first place? Defend the model? What are its advantages? You might comment on its suitability from a theoretical point of view as well as indicate practical reasons for using it. Provide a rationale. State your specific objectives, and describe the reasoning that led you to select them. Very briefly describe the research design
and how it accomplished the stated objectives.
Authors’ Response: Thank you for your insightful suggestion and feedback. We carefully reviewed and revised overall manuscript based on your and the other reviewers’ comments.
R1 Comment 4: Please distinguish your work from other work on the subject, and highlight how you extend such work to make a contribution to electronic commerce and practice.
Be sure to distinguish between your contribution and prior/existing work.
Authors’ Response: We reflected your comments in our revision and made the change.
[see P. 4. Line 143-145] Consequently, because different ways of using social media and their impact on body satisfaction likely differ by culture, fitness apparel companies would do well to diversify, by culture, their communications about products closely tied to body satisfaction.
R1 Comment 5: Please cite more of the Sustainability papers if you are submitting manuscripts to this journal.
Authors’ Response: We cited more recent articles (e.g., Hossain, M., & Kim, M. Does Multidimensional Service Quality Generate Sustainable Use Intention for Facebook?. Sustainability. 2018, 10(7), 2283)
Reviewer 2 Report
This research article provides a comparison between the United States and South Korea. However the authors often refer to the generic term of ”Korean consumers”. It is important to make a clear distinction between South Korea and North Korea because there are two distinct states.
The authors must explain clearly and in detail why the two states were selected, i.e United States and South Korea. What are the common points, but the differences between these two countries so that the analysis to be relevant? For example, both selected countries are developed economies according to international criteria such as FTSE Country Classification listed on September 2019. How important is the level of development for a sustainable approach?
The aim of the work should be more highlighted.
The contribution to existing knowledge is average.
I recommend updating the bibliography with recent and relevant sources from 2018 and 2019.
The framework regarding the sustainability concept must be considerably extended and correlated with the subject analysed. The research paper includes very few ideas regarding sustainable growth. A significant improvement in this regard is required.
Author Response
R2 Comment 1: This research article provides a comparison between the United States and South Korea. However the authors often refer to the generic term of ”Korean consumers”. It is important to make a clear distinction between South Korea and North Korea because there are two distinct states.
Authors’ Response: We appreciate your suggestion and feedback. We made a change for the word “Korea” to “South Korea” in the entire manuscript.
R2 Comment 2: The authors must explain clearly and in detail why the two states were selected, i.e United States and South Korea. What are the common points, but the differences between these two countries so that the analysis to be relevant? For example, both selected countries are developed economies according to international criteria such as FTSE Country Classification listed on September 2019. How important is the level of development for a sustainable
approach?
Authors’ Response: Suggestion was well received and we made a change for introduction
[see P. 2. Line 82-85] We can thus infer that while e-shopping and social media are widely popular in both South Korea and the U.S., the two countries have contrasting cultural backgrounds due to the different values and beliefs associated with Eastern and Western cultures.
R2 Comment 3: The aim of the work should be more highlighted.
Authors’ Response: We reflected your comments in our revision and made the change in purpose of this study.
[see P. 3. Line 92-95] Therefore, the purpose of this study was to investigate (1) comparison of college students’ body image satisfaction in both the United States (U.S.) and South Korea and (2) how their body satisfaction influences consumer communication and the sustained referral intentions of fitness apparel in social media.
R2 Comment 4: I recommend updating the bibliography with recent and relevant sources from 2018 and 2019.
Authors’ Response: we thoroughly updated several citations, which are necessary in our manuscript based on your comments.
R2 Comment 6: The framework regarding the sustainability concept must be considerably extended and correlated with the subject analyses. The research paper includes very few ideas regarding sustainable growth. A significant improvement in this regard is required.
Authors’ Response: Suggestion was well received and we made a change for discussion and implication [see P.13. Line 453-460]. What is also important to take away is that our conceptual model serves to identify the effect of social capital on sustained WOM. Moreover, our results reveal that social capital influences sustained referral intention in social media and e-marketing, with online service quality a mediating factor. In other words, social capital impacts e-service quality perception, and perceived e-service quality perception impacts WOM. Accordingly, we suggest that, in order to sustainably grow, online retailers would do well to link SNS with their homepages for reviews and recommendations—any good recommendations of a product strengthen its producer’s social capital [70], a principle that holds true online as well [71].
Reviewer 3 Report
It would be useful to insert at the end of the Introduction what the purpose of the study is, what methodology it is intended to apply, and the two questions formulated to become objectives.
The expression "Intercultural differences" is supported by two atypical variables: Social Media (a channel of communication) and satisfaction (degree of satisfaction).
I recommend reading the actual studies by Geert Hofstede to understand correctly what the "intercultural differences" entail. You can then extract dimensions such as: explicit-implicit or egalitarian cultures - hierarchical cultures.
The variables that can be used in the case of cultural differences can be: degree of insecurity; prejudices; stereotypes; eating habits; individualism - collectivism ...
It is better to insist on Literature Review.
The second question is unclear; I recommend to reformulate it.
To be reformulated H1.1. What is specific to the US? What is specific about Korea?
To be reformulated H1.2. - Contains 3 aspects in one sentence. What do you really want?
There is no null hypothesis.
Author Response
R3 Comment 1: It would be useful to insert at the end of the Introduction what the purpose of the study is, what methodology it is intended to apply, and the two questions formulated to become objectives.
Authors’ Response: We reflected your comments in our revision and made the change in purpose of this study. [see P.2-3. Line 93-95] The purpose of this study was to investigate (1) comparison of college students’ body image satisfaction in both the United States (U.S.) and South Korea and (2) how their body satisfaction influences consumer communication and the sustained referral intentions of fitness apparel in social media.
R3 Comment 2: The expression "Intercultural differences" is supported by two atypical variables: Social Media (a channel of communication) and satisfaction (degree of satisfaction).
I recommend reading the actual studies by Geert Hofstede to understand correctly what the "intercultural differences" entail. You can then extract dimensions such as: explicit-implicit or egalitarian cultures - hierarchical cultures.
Authors’ Response: We appreciate your thoughtful comment again. These comments were very helpful, and we thoroughly revised our literature section and went over and removed or added a few sentences in our theoretical background and research hypotheses.
R3 Comment 3: The variables that can be used in the case of cultural differences can be: degree of insecurity; prejudices; stereotypes; eating habits; individualism - collectivism ...
It is better to insist on Literature Review.
Authors’ Response: We revised sentences and updated a few citations, which are necessary in our literature reviews [see P.3. Line 124-145].
R3 Comment 4: The second question is unclear; I recommend to reformulate it.
To be reformulated H1.1. What is specific to the US? What is specific about Korea?
To be reformulated H1.2. - Contains 3 aspects in one sentence. What do you really want?
There is no null hypothesis.
Authors’ Response: We reflected your comments in our revision and made the change in our hypotheses H1 to H5.
[see P. 5. Line 187-191]
H1: Social capital in SNSs positively influences perceived e-service quality in terms of website design.
H2: Social capital in SNS positively influences perceived e-service quality in terms of website responsiveness.
H3: Social capital positively influences sustained WOM for fitness apparel.
[see P. 5. Line 227-230]
H4: Perceived e-service quality in terms of website design influences sustained WOM for purchasing fitness apparel.
H5: Perceived e-service quality in terms of website responsiveness influences sustained WOM for purchasing fitness apparel.
Round 2
Reviewer 2 Report
The article has been significantly improved and can be considered for publication.
Reviewer 3 Report
I appreciate that you have taken into account the suggestions I have made and you have improved the material. From my point of view it can be published.